# Brain-Inspired Data Transmission in Dense Wireless Network

**DOI:** 10.3390/s21020576

**Published:** 2021-01-15

**Authors:** Łukasz Kułacz, Adrian Kliks

**Affiliations:** Institute of Radiocommunications, Poznan University of Technology, 61-131 Poznan, Poland; adrian.kliks@put.poznan.pl

**Keywords:** dense wireless network, percolation threshold, reliability, stochastic geometry

## Abstract

In this paper, the authors investigate the innovative concept of a dense wireless network supported by additional functionalities inspired by the human nervous system. The nervous system controls the entire human body due to reliable and energetically effective signal transmission. Among the structure and modes of operation of such an ultra-dense network of neurons and glial cells, the authors selected the most worthwhile when planning a dense wireless network. These ideas were captured, modeled in the context of wireless data transmission. The performance of such an approach have been analyzed in two ways, first, the theoretic limits of such an approach has been derived based on the stochastic geometry, in particular—based on the percolation theory. Additionally, computer experiments have been carried out to verify the performance of the proposed transmission schemes in four simulation scenarios. Achieved results showed the prospective improvement of the reliability of the wireless networks while applying proposed bio-inspired solutions and keeping the transmission extremely simple.

## 1. Introduction

The current model of life, the development of technology, and access to different services increase the demand for access to spectrum. The interest in wireless access to an ever wider range of services is constantly growing. In view of the limited radio resources and problems related to the use of subsequent frequency bands, the solution to this problem could be sought in improving the efficiency of spectrum access. Observing the trends and plans related to the fifth generation of wireless cellular networks and beyond, one can notice—among other things—the continuous increase of the device’s density per certain area in the network [1,2]. The proliferation of smartphones and other wirelessly connected devices in the community is greater from one year to another. However, at the same time, the network infrastructure becomes continuously more heterogeneous—services offered by the macro-base stations are supported by densely deployed micro- and pico-base [3,4]. Moreover, the cloud-based solutions gain popularity in practice, where the centralized baseband unit (BBU) associated with a number of connected remote heads (RH) [5]. One of the justifications for this situation is the fact that the reduction of distance between the network edge and the end-user leads toward better transmission opportunities. It is expected that the number of wireless nodes (mainly infrastructure elements like base stations, repeaters, etc.) will increase also in the future. Thus, in our paper we consider ultra-dense wireless networks [6,7], where the density of transmitting nodes per certain area is as huge that in most cases line-of-sight transmission is possible, and in consequence, the resultant transmit power at each hop may be significantly reduced compared to the contemporary nodes. Moreover, following the recent concept of so-called integrated-access-backhaul (IAB) transmitters, we assume that each wireless nodes may act relay [8]. In consequence, in our work, we focus on ultra-dense wireless networks, where each node may act as a source, sink, or relay in the network. Such an approach brings immediately to mind the association with human brain and human nervous system (HNS) [9,10,11,12]. Observation of the operation of the human body, in particular how information is transmitted in the body (which makes us move our hand, see, feel) seems to be highly similar to such ultra-dense telecommunication networks. Many elements of the human body are interconnected and exchange information with each other. A very important observation is that this information exchange takes place in the nervous system, which consists mainly of a very dense network of neurons. Additionally, this transmission is very simple yet complete. This is reminiscent of the notion that nature has developed the nervous system in such a way that information is transmitted by simple transmission over a large number of (independent/separate) connections instead of complicated transmission over a small number of paths (usually common). Inspired by such observation, we have considered the idea of a highly simplified data transmission scheme, where the goal is to guarantee the fulfillment of various network performance criteria (such as network reliability) while minimizing the burden due to the information processing and delivery, which should be as simplified as possible. In our prior work we tried to mimic the behavior of such dense wireless networks by applying the mechanisms found in the human brain and nervous system, e.g., [13]. In particular, in addition to the transmit-receive nodes, the presence of the specialized devices is considered that support self-organization and self-healing of the system, so in consequence its failure-free functioning.

However, in today’s network deployment the reliability plays an extremely important role. The messages have to be delivered correctly with a very low level of possible transmission failures [14,15]. In order to correctly assess the reliability of the proposed solutions, deterministic methods such as Fault Trees, Reliability Block Diagrams (RBD), or Markov chains are often used [13]. Unfortunately, in the case of dense networks with a significant number of nodes, the problem of the high computational complexity of these methods becomes crucial. For this reason, the authors used stochastic geometry [16,17], or more precisely the percolation theory, to assess the reliability of the dense wireless system. Particularly noteworthy here is the percolation threshold, which can be used to determine the critical number of nodes necessary for the correct operation of the assumed network. Initial results have been presented in [18,19,20], where in [18] the percolation was applied to evaluate the reliability of wireless network under different nodes density, in [19] the impact of microglia inspired nodes were tested in case of distributed antenna systems, and in [20] network nodes was evaluated in terms of reliability in the case of the limited lifetime of nodes. All mentioned initial works use an idealistic transmission model which omits a very important aspect in case of a dense wireless network - interference. In this research, we extend this work by including an analysis of interference impact on reliability.

Moreover, in our prior works, we have applied two research methodologies to evaluate the performance of the biology-inspired dense wireless networks. In particular, we have adjusted the classic LOADng routing algorithm [21] and implemented it in the considered dense wireless transport network. In this case, four different types of HNS-based inspirations for dense wireless networks have been tested. Next, the theoretical analysis using percolation theory has been applied and verified by means of extensive computer simulations. However, in none of our prior work, we have evaluated the real impact of mutual (self-)interference on the overall performance of the system. As we consider a fully distributed system (i.e., the one with no central coordination), the probability of simultaneous data transmission by many nodes located in their close vicinity may be high. Thus, it is necessary to consider the presence of such phenomena. The main contribution of this paper is the evaluation of the impact of interference on the system performance. We did it in two ways, first, we verify the impact of interference in such a dense wireless systems by application of the percolation theory, and second, we carry out extensive computer simulations, where the LOADng algorithm is modified by adding a back-off procedure. In a nutshell, the novel contributions of this paper compared to all prior works, are the following:in this paper, we have performed the final simulations where all previous findings have been jointly included in one computer experimentsnext, the detailed analysis of the self-interference between the nodes has been investigateda mathematical analysis of the percolation threshold for the dense wireless network has been providedthe LOADng routing algorithm has been significantly improved to deal with self-interference.

The structure of the manuscript is the following. First, we briefly overview the HNS and how it has inspired us in proposing new modifications in dense wireless networks. Next, we present our system model and scenarios selected for investigation and discuss the research problem, i.e., the evaluation of the expected impact of self-interference phenomena. The next section deals with the reliability analysis, including percolation theory and its implementation in dense wireless networks. Achieved simulation results are then discussed in the following section, and finally, the paper is concluded.
In this paper, we perform the final simulations where all previous findings are jointly included in one computer experiments.A detailed analysis of the self-interference between the nodes is investigated.A mathematical analysis of the percolation threshold for the dense wireless network is provided.The LOADng routing algorithm is significantly improved to deal with self-interference.

The structure of the manuscript is the following. First, we briefly overview the HNS and how it inspired us in proposing new modifications in dense wireless networks. Next, we present our system model and scenarios selected for investigation and discuss the research problem, i.e., the evaluation of the expected impact of self-interference phenomena. The next section deals with the reliability analysis, including percolation theory and its implementation in dense wireless networks. Achieved simulation results are then discussed in the following section, and, finally, the paper is concluded.

## 2. Related Work

Humans have been always highly inspired by the functioning of our world and nature, and in particular of the human body; these inspirations lead to numerous significant findings and discoveries. In the context of computer science, there are various algorithms and solutions which are motivated by the observation of nature. One can mention for example solutions based on the behavior of bees, bumblebees, or ant-colony, in general, swarm intelligence [22,23,24], an algorithm based on findings in genetics, but also solutions like simulated annealing which mimics the behavior of the fabric processing. Recently, the most popular solutions are related to the application of the neural networks in their various configurations (like convolution neural networks, recursive neural networks, etc.), and implementation contexts (computing, feature recognition, automation, networking) [25,26,27]. It is also worth mentioning that the behavior of bees (swarm) has inspired the design of the whole communication and networking frameworks and protocols, like ZigBee [28]. However, when projecting the brain-inspirations on wireless communications, the number of publications is not that high, which suggests that this domain of research is still unexplored. However, it is worth mentioning some works, such as [29], where a special class of neuromorphic computing (mainly reservoir computing) is applied for signal detection in multiple input multiple output orthogonal frequency division multiplexing systems. In [30], the authors have proposed the brain-inspired method for constructing a robust virtual wireless sensor network. Next, in [31], the dynamic spectrum management in cognitive wireless ad-hoc networks was investigated. A very promising approach has been presented in [32], where the authors proposed the neural wireless sensor networks inspired by the functioning of the neural network.

Recently, increased interest is also observed in so-called bio-molecular communications, as presented in [33]. Synaptic transmission and neuro-spike transmission have been analyzed in detail in the context of communications in [34,35,36].

Furthermore, the authors in [37], proposed to use the property of neurons to connect to many other neurons (both as an information source and an information outlet) in sensor networks, where a dynamic node grouping mechanism is proposed. In turn, in [38] one can find an algorithm for selecting nodes and bit rates in sensor networks which bases on the immune system.

## 3. Ultra-Dense Wireless Networks and Their Biological Counterparts

In our work, we consider a highly dense network consisting of *N* wireless nodes deployed randomly over a certain square area of size *D*, as illustrated in Figure 1. Each wireless network node may act as a transmitter (source), a receiver (sink), or as a relay of the message. In other words, the network of all wireless nodes may be treated as some sort of transport network. These wireless nodes operate in the same frequency band (central frequency $f_{c}$ and bandwidth *B*); thus, when transmitting at the same time, they may cause mutual interference. Moreover, we assume that any two wireless nodes which are close enough to receive the signal with the power above some threshold create the wireless link.

The illustrative situation is shown in Figure 1, where the light-green circle represents the coverage area of the transmitting node and the common part of two such areas is marked by orange and represents the interference area. All nodes within the interference area are susceptible to harmful interference and degradation of the signal-to-interference-plus-noise metric. Without loss of generality, in our further analysis, we select one of the nodes as transmitter and the other distant node as the destination node for this reception of this message; the other nodes laying between them are treated as relays. As the network is dense, we follow our prior inspirations from the human nervous system (HNS), which we recap briefly below.

### 3.1. Fundamentals of Human Nervous System

The human nervous system consists of nerve and glial cells. They form a very dense network of nerve connections that transmit information. This information reflects the data that control the entire human body: simple information about touch, complex data about the environment (vision and sound), muscle control, thinking, imagination, etc. The basic transmission of impulses takes place by means of neurons which, using both electrical and chemical machinery, pass the information on. Glial cells are generally designed to support the transmission that takes place between neurons. Despite the fact that neurons do not physically touch—there is a non-zero distance between them—this transmission is closer to wired transmission than wireless transmission. The neuron, after receiving enough neurotransmitters via dendrites, creates/excites an action potential in the cell body, and then, as an electrical signal, it is sent along the axon. At the end of the axon are synapses which, upon receiving a sufficiently strong electrical impulse from the axon, produce neurotransmitters. These, in turn, released from a given neuron, are designed to excite (in an analogous way) another neuron. Much of the impulse path is thus traveled in the form of an electrical impulse in the axon (which is analogous to wired data transmission). The information transmission mechanism itself, however, is very similar to wireless transmission between devices in a network, e.g., sensor or relay network.

Despite being neurons, there are different types of glial cells:Astrocytes are the glial cells responsible for constituting the so-called blood–brain barrier (i.e., the barrier between the human nervous and blood systems) [9]. The astrocyte is able to detect the presence of some chemicals in the blood (such glucose) through dedicated receptors, and in consequence influence the transmission between neurons.Microglia are also highly important glial cells in HNS being the first and main form of human active immune defense in the central nervous system. In a nutshell, microglia clean the central part of HNS from unnecessary and damaged neurons or cells. Microglia cells may act in two states: observation of the surrounding and taking repairing action. It is worth emphasizing that microglia can communicate with other microglia cells, astrocytes or even nerves [9].Oligodendrocyte and Schwann cells are the glial cells responsible for the creation of the myelin sheath on axons, thus providing insulation for axons, and the process of creating a myelin sheath is called myelination. We can find oligodendrocytes in the central nervous system, and this cell wraps up to 50 axons with myelin. In turn, Schwann cells can be found in the peripheral nervous system, and they wrap only one axon with myelin. Myelination is an important part of human intelligence; it may be treated as a learning procedure or our nervous system development [9].

There are other glial cells in the human body, but, in our research, we considered the application of two kinds of wireless nodes inspired directly by functionalities provided by astrocytes and microglia.

### 3.2. Inspirations for Wireless Network

In our research, we considered the application of the following, bio-inspired functionalities of the dense wireless system:Neurons: They may be treated as the rudimental kind of wireless node, i.e., a simple transmission–reception point in a wireless network.They may possess capabilities to receive (neurotransmitters in a neuron; radio waves reception in a wireless network), process (cell body in a neuron; a processor in transmission point), and transmit (axons and synapses in a neuron; radio waves emission in a wireless network). In HNS, a neuron will be fired (i.e., it will pass on the nervous pulse), if the incoming pulse from the preceding neuron is strong enough. In the context of wireless communication, such a scheme may be mapped to the case where the wireless transceiver relays the message only of the received signal power is above some predefined threshold. For further investigation, in our dense wireless system, such transceivers are marked as neuron-nodes.Astrocyte: Astrocyte function can be embodied in wireless communications as the connection point between different kinds of parts of the network, i.e., some parameters of one network can influence the functionality of another network through the astrocytes. A practical example could be the presence in the wireless network of some dedicated sensors deployed for the detection of various emergency situations. Once detected, the network can react by, e.g., redirecting data traffic omitting unstable parts of the network. In other words, this function can be implemented as a short-term scale control plane that can reconfigure some part of the network in sudden situations. For further investigation, in our dense wireless system, such transceivers are marked as astrocyte-nodes.Microglia: These glial cells may be seen as nodes providing some fault tolerance to the network. It could react in several ways, for example, it may inform other devices or some network controller about the identified problem. Moreover, it could take over tasks that were done by the currently-not-responding device. Microglia functionality in this case is very similar to astrocyte functionality. However, the microglia inspired device is part of the transmission network, whereas the astrocyte-like device can be treated as an external input to the network from other networks. For further investigation, in our dense wireless system, such transceivers are marked as microglia-nodes.Oligodendrocyte: Myelination could be understood as a network learning capability. This function can be simply implemented as a long-term scale control plane.

### 3.3. The Bio-Inspired Vision of the Dense Wireless Network

As mentioned previously, we consider a highly dense wireless network, where each wireless node may act as a relay. Although the mechanisms known from HNS have been already widely in communications, it was done mainly in the context of wired connections. Moreover, inspirations of neural networks are successfully applied in practice as a highly efficient and reliable tools in the artificial intelligence domain. However, the main idea of the authors was to project the nervous system onto a wireless telecommunications network, which is not that straightforward [13,18]. Summarizing our prior findings, we assumed the following relations, which specify the rationale of our proposed highly simplified transmission scheme:This idea treats a single network node as a neuron, thus simplifies its functions to a minimum, i.e., to the simple transfer of information. Consequently, no channel coding is applied, and no retransmission is allowed in the normal functioning of the network. The transmit power is reduced to the required minimum, and the message is treated as delivered correctly when the required signal power (over noise and interference) is observed at the reception node. In detail, we assume the required threshold for the signal-to-interference-plus-noise (SINR) level, necessary to demodulate transmitted signals. Furthermore, typically, the simplest modulation schemes are considered (e.g., binary or quadrature phase-shift keying, denoted as BPSK and QPSK, respectively).The proposed network includes additional devices that reflect the functioning of the microglia in the neural system. These are redundant devices that do not take part in the transmission and are designed to support the transmission that takes place between the basic network nodes (neurons).The transmission within the network is not controlled centrally. Moreover, the nodes transmit their messages immediately when the message is ready for further transmission. Consequently, it may happen that many signal transmission may be initiated by various nodes deployed in a certain area resulting in performance degradation of the network due to increased interference power.

One may observe that the above assumptions relax numerous solutions applied typically in contemporary wireless systems, such as advanced channel coding. They simplify the processing (and finally energy consumption) in the network but may result in deterioration of the overall system performance. Thus, it is mandatory to verify whether the presented inspirations guarantee the required level of performance of the wireless transmission system. Thus, the ultimate goal of our study is to consider a highly simplified transmission scheme and check how it influences the system reliability.

### 3.4. Node-Types and Investigation Scenarios

Following the above discussion, three types of nodes are considered in the further evaluation, mainly rudimental wireless relaying nodes possessing the neuron-functionality. These neuron-nodes are capable of message passing to the neighboring nodes only if the received signal strength is above the specified threshold (one may think of node activation as an analogy to neuron-cell firing). Next, there are microglia-nodes that do not take any action in message transmission until at least one of the neighboring neurons stops working properly. In such a situation, the microglia-node may detect such a situation, and then turn on its transmission functionality and act as a neuron-node. Finally, an astrocyte-node has the specific functionality to collect and process data about some problems in its vicinity. The astrocyte-nodes leads to, e.g., bypassing the less-reliable area by finding other paths in the network or by leaping it over by increasing the transmit power.

### 3.5. Related Work

Humans have always been highly inspired by the functioning of our world and nature, and in particular of the human body; these inspirations lead to numerous significant findings and discoveries. In the context of computer science, there are various algorithms and solutions which are motivated by the observation of nature. One can mention for example solutions based on the behavior of bees, bumblebees, or ant-colony, in general, swarm intelligence [22,23,24], an algorithm based on findings in genetics, but also solutions such as simulated annealing which mimics the behavior of the fabric processing. Recently, the most popular solutions are related to the application of the neural networks in their various configurations (e.g., convolution neural networks, recursive neural networks, etc.) and implementation contexts (computing, feature recognition, automation, and networking) [25,26,27]. It is also worth mentioning that the behavior of bees (swarm) has inspired the design of the whole communication and networking frameworks and protocols, e.g., ZigBee [28]. However, when projecting the brain-inspirations on wireless communications, the number of publications is not that high, which suggests that this domain of research is still unexplored. However, it is worth mentioning some works, e.g., [29] applied a special class of neuromorphic computing (mainly reservoir computing) for signal detection in multiple input multiple output orthogonal frequency division multiplexing systems. In [30], the authors proposed the brain-inspired method for constructing a robust virtual wireless sensor network. Next, in [31], the dynamic spectrum management in cognitive wireless ad-hoc networks is investigated. A very promising approach is presented in [32], where the authors proposed the neural wireless sensor networks inspired by the functioning of the neural network.

Recently, increased interest is also observed in so-called bio-molecular communications, as presented in [33]. Synaptic transmission and neuro-spike transmission are analyzed in detail in the context of communications in [34,35,36].

Furthermore, the authors is [37] proposed to use the property of neurons to connect to many other neurons (as both an information source and an information outlet) in sensor networks, where a dynamic node grouping mechanism is proposed. In turn, in [38], one can find an algorithm for selecting nodes and bit rates in sensor networks which is based on the immune system.

## 4. Network Reliability Assessment through the Percolation Theory

The performance and efficiency of communication networks are expressed in terms of various metrics. One of them, seriously considered in recent years in terms of evaluation of future wireless networks, is reliability. In the context of the fourth generation of the cellular networks, the level of so-called “four nines” was considered as the target of guaranteed network reliability. Analogously, “five nines” have been treated as the goal for the fifth generation. These values correspond in general to the probability of lack of packet delivery at the levels of 0.01% and 0.001%, respectively. When looking at the pure definition of the term, reliability is the probability specifying that the wireless network operates properly without any device or link failure [39]. However, these terms should not be confused with availability, which describes the proportion of time when the system is in a functioning condition. Various methods can be used for reliability assessment, e.g., Markov Chains, Fault Trees, or RBD [39]. However, these approaches do not reflect well or easily the stochastic nature of such dense wireless networks, whose performance strongly depends on the current node distribution and node load.

### 4.1. Application of the Percolation Theory

Stochastic geometry allows modeling and interpreting the impact of various processes and the random variables characterized by various spatial distributions. In other words, the impact of randomness in the spatial domain can be evaluated. Among various tools known from stochastic geometry, percolation theory allows specifying the value of the probability when the network will become blocked (no path may be identified between the source and the destination) for a given network structure (i.e., spatial distribution of nodes) [40,41,42].

As a physical phenomenon, percolation describes the mechanism by which porous material is permeated. A very common example here may be the coffee brewing process, where the water flows through coffee beans to extract the appropriate ingredients. Depending on the type of coffee and its amount (densely or loosely grounded coffee, amount of coffee, etc.), the percolation process will run in a different way. One may apply percolation theory to describe this process, where the liquid flow is defined by the path in the graph, and the graph represents the material (coffee in this example). In the case of wireless networking, the theory of percolation allows analyzing the message passage through networks with various densities and spatial distributions, which we can call permeability. In that context, one may observe that for a given material (network structure) there exists a critical parameter called percolation threshold, which specifies the limit, and exceeding this limit informs about the lack of permeability. In particular, the percolation threshold defines the percentage of nodes that must be active to guarantee permeability. Thus, for a given set of nodes *N* and the achieved percolation threshold $p_{T}$, the minimum required a number of nodes that guarantee the proper functioning of the network is equal to $\left\lfloor p_{T}\cdot N\right\rfloor$, where ⌊·⌋ is the floor function.

Let us assume that the wireless network is represented in form of a graph, where the wireless transceiver corresponds to the node and the valid wireless link to the graph edge. Assuming some fixed maximum transmit power, the corresponding transmission range will be achieved. One may observe that, for the fixed range, when the density of nodes decreases, the probability of lack of permeability decreases as well. Analogous observations can be made when reducing the effective transmission range (due to, e.g., reduced transmit power or increased observed interference at the reception node). If the permeability is not achieved, the network graph contains some clusters (subgraphs). Following the authors of [40,41,42], the percolation threshold is reached at the moment when the size of the second-largest cluster (denoted hereafter as SG) in the graph reaches its maximum.

Following [18], we derive the percolation threshold for the dense wireless node to show the reliability of the bio-inspired communications schemes, however, in this paper we consider a significant aspect, mainly the presence of self-interference, as illustrated in Figure 1. We assume that the network nodes are distributed according to the spatial Poisson process and are operating asynchronously. In our simplified case, we state that there exists a reliable wireless link between two wireless nodes, when the observed signal-to-interference-plus-noise ratio SINR is above some threshold $\theta^{*}$, i.e., $SINR\geq \theta^{*}$. As in [42], the desired signal may be expressed as $S=\frac{P_{0}\prod_{k}X_{k}}{R^{2b}}$, where $X_{k}$ represent the existing propagation effects in the wireless channel (such as path loss $X_{k}=1$ or log-normal fading $X_{k}=e^{2\sigma G}$, for *G* being the normally distributed random variable of zero mean and unit variance). Moreover, *R* is the distance and *b* amplitude loss exponent. Analogously, the total sum-interference from all j=1,2,…,J simultaneously transmitting nodes can be computed as $I=\sum_{j=1}^{J}\frac{P_{I}\Delta_{j}\prod_{k}X_{j,k}}{R_{j}^{2b}}$, where $X_{j,k}$ represents propagation effects in link to device *j*. In our case we assume that the network is not synchronized, thus, in order to describe the transmission profile (i.e., the duty cycle) we introduce the random variable Δ∈[0,1]. Thus, the total aggregated interference *I* can be defined utilizing the skewed stable distribution [42]:(1)$$I∼S\left(\alpha =\frac{1}{b},\beta =1,\gamma =\pi \lambda C_{1/b}^{-1}P_{I}^{1/b}E\left\{\Delta_{j}^{1/b}\right\}\prod_{k}E\left\{X_{j,k}^{1/b}\right\}\right).$$
where $\Delta_{j}$ is the duty-cycle factor associated with interferer *j*.

Here, in this generic formula, the term $C_{\alpha }$ is defined as
(2)$$C_{\alpha }=\left\{\begin{array}{cc} \frac{1-\alpha }{\Gamma (2-\alpha )\cos\left(\pi \alpha /2\right)}, & \alpha \neq 1\\ 2/\pi , & \alpha =1 \end{array}\right.,$$
where Γ() denotes the gamma function. Next, the $P_{I}$ term stands for the transmit power of the interfering nodes. In our case, however, we assume that the maximum power is the same for the desired node and the interferers. The introduced skewed distribution S(α,β,γ), valid for α∈(0,2], β∈[−1,1] and γ≥0, can be characterized by its skewness β, and it possesses the corresponding characteristic function
(3)$$\omega \left(w\right)=\left\{\begin{array}{cc} \exp\left(-\gamma {\left|w\right|}^{\alpha }\left[1-j\beta \mathrm{sign}\left(w\right)\tan\left(\frac{\pi \alpha }{2}\right)\right]\right), & \alpha \neq 1\\ \exp\left(-\gamma \left|w\right|\left[1+j\frac{2}{\pi }\beta \mathrm{sign}\left(w\right)\ln\left(w\right)\right]\right), & \alpha =1 \end{array}\right.$$

It is well known that the probability density function of our searched random variable *I* can be calculated using inverse Fourier transformation
(4)$$f\left(x\right)=\frac{1}{2\pi }\int_{-\infty }^{\infty }e^{-isx}\omega \left(s\right)ds.$$

In our case, we assume that the nodes are densely deployed over a certain area, thus the dominant channel effect will be path loss at the line-of-sight link, i.e., $X_{k}=1$. Again, following the authors of [42], the probability of SINR above-assumed threshold $\theta^{*}$ can be derived as
(5)$$P\left\{SINR\geq \theta^{*}\right\}=F_{I}\left(\frac{P_{0}}{r_{0}^{2b}\theta^{*}}-\sigma_{n}^{2}\right),$$
where $\sigma_{n}^{2}$ is a noise level, $r_{0}$ is a distance between nodes, *b* is a propagation environment coefficient, $P_{0}$ is a transmit power of the desired signal, and $F_{I}$ is a cumulative distribution function of random variable *I*. Consequently, Formula (Equation 1) can be simplified as
(6)$$I∼S\left(\alpha =\frac{1}{b},\beta =1,\gamma =\pi \lambda C_{1/b}^{-1}P_{I}^{1/b}E\left\{\Delta_{i}^{1/b}\right\}\right)$$
where α, β, and γ are characteristic function coefficients. Consequently, the corresponding characteristic function can be derived as
(7)$$\omega \left(w\right)=\left\{\begin{array}{cc} \exp\left(-\gamma {\left|w\right|}^{\frac{1}{b}}\left[1-j\mathrm{sign}\left(w\right)\tan\left(\frac{\pi }{2b}\right)\right]\right), & b\neq 1(\mathrm{non}\mathrm{line}\mathrm{of}\mathrm{sight})\\ \exp\left(-\gamma \left|w\right|\left[1+j\frac{2}{\pi }\mathrm{sign}\left(w\right)\ln\left(w\right)\right]\right), & b=1(\mathrm{line}\mathrm{of}\mathrm{sight}). \end{array}\right.$$

In our highly dense wireless network, we assume the presence of line-of-sight, and the above formula for b=1 is selected. Moreover, the value of γ is defined as:(8)$$\gamma =\pi \lambda C_{1}^{-1}P_{I}E\left\{\Delta_{i}\right\}=\frac{\pi^{2}}{2}\lambda P_{I}E\left\{\Delta_{i}\right\}$$
as $C_{1}^{-1}=\frac{2}{\pi }$. Thus, the distribution of the aggregated interference can be expressed as:(9)$$I∼S\left(\alpha =1,\beta =1,\gamma =\frac{\pi^{2}}{2}\lambda P_{I}E\left\{\Delta_{i}\right\}\right).$$Finally, the probability density function can be computed as:(10)$$\begin{array}{cc} f\left(x\right) & =\frac{1}{2\pi }\int_{-\infty }^{\infty }e^{-iwx}\omega \left(w\right)dw\\ \; & =\frac{1}{2\pi }\int_{-\infty }^{\infty }e^{-iwx}e^{-\gamma \left|w\right|\left[1+j\frac{2}{\pi }\mathrm{sign}\left(w\right)\ln\left(w\right)\right]}dw\\ \; & =\frac{1}{2\pi }\int_{-\infty }^{\infty }e^{-iwx}e^{-\frac{\pi^{2}}{2}\lambda P_{I}E\left\{\Delta_{i}\right\}\left|w\right|\left[1+j\frac{2}{\pi }\mathrm{sign}\left(w\right)\ln\left(w\right)\right]}dw. \end{array}$$

The above formula can be calculated numerically and tabulated, allowing for direct calculation of the searched probability $P\left\{SINR\geq \theta^{*}\right\}$.

### 4.2. Experiment Setup

To evaluate the investigated concept of highly simplified transmission in dense wireless networks, we conducted extensive computer experiments. The whole simulator was prepared using Matlab version 9.4.0.813654 (R2018a) with standard libraries, and all simulation runs were carried on the standard personal computer PC with specification i7-6500u, 12 GB RAM, under Windows 10 system. In our experiment analysis, we apply normalized values to generalize it and make it independent from the applied physical values. Thus, the analyzed system consists of a network of $N=\left\{30\right\}$ randomly distributed neuron-nodes. These devices mirror neurons in the nervous system, i.e., they are designed to transmit information to each other. Moreover, once the neuron-nodes are deployed, M=0.1N microglia-nodes are added, deployed also uniformly in the same area. (Please note that in this investigation phase we intentionally do not deploy astrocyte-nodes.) The analyzed area is a square with the side D=10. The devices are characterized by the transmitting power $P_{0}=P_{I}$ and the range r=1. The required signal to noise and interference level was set at $\theta^{*}=1$ (in a linear scale) and the noise level at $\sigma_{n}^{2}=1$. All simulation parameters used in this part of our work are summarized in Table 1.

During the experiment, we assume that the simulations searches for the percolation threshold value that ensures correct operation of the network. The simulations were averaged (repeated) over different topologies and different device operation probability levels. An example of the analyzed topology is shown in Figure 2. One may observe the presence of wireless nodes (corresponding to neurons, denoted by black dots), the source, and the destination.

In the following results, we show the size of the SG cluster as the function of the probability $p_{c}$, which is the probability of proper functioning of any wireless node. Each simulation result is presented as a solid line and the additional dashed line is a fourth-degree polynomial approximation. As mentioned above, the greatest value of SG is achieved at the percolation threshold, i.e., $p_{T}=\arg\max_{p_{c}}SG\left(p_{c}\right)$. The smaller is the $p_{T}$, the better is the reliability of the network (i.e., fewer nodes are required to guarantee permeability). In other words, when the percolation threshold decreases, the network will be able to operate in heavier conditions (with greater error probability at a certain link). In Figure 3 and Figure 4, one may observe the changes of SG as function of different transmit power levels and for different number of neuron-nodes. One may observe that when the transmit power increases (so the coverage area increases as well, and, in consequence, for the fixed node density the number of nodes located in interference areas increases) the percolation threshold slightly shifts left, but the minimum is achieved for $p_{c}\approx 0.25$. However, the change in node density not only shifts the curves left, but also emphasizes the effect that for higher transmit power the percolation threshold also decreases. In Figure 5, one may observe the changes of SG as function of different number of nodes. These results are mainly two selected lines in Figure 3 and Figure 4 for the same level of power $P_{0}=P_{I}=30$ but for different number of nodes, i.e., N={30,100}.

Finally, in Figure 6, the impact of the duty cycle on the network reliability is shown. Interestingly, as the size of the SG changes with the change of the mean duty cycle, the network reliability (expressed by means of the percolation threshold) is rather unchanged (around 0.2).

In this section, we derive theoretically how the percolation threshold can be derived in the dense wireless network, as discussed in details in [42]. However, the percolation threshold provides only some insights toward the overall theoretic limits of the dense wireless network. Thus, in the following section, we investigate the network reliability utilizing the extensive computer simulations.

## 5. Bio-Inspired Wireless Network Reliability—Analysis by Means of Computer Experiment

As the previous section discussed some theoretic limits for the reliability of the dense wireless network, in this part we analyze it by means of computer simulations and experiments. We have extended the prior works, by incorporating the self-interference in the research, and thus, this section describes the results of the simulation in which the proposed concepts were tested in the network where the messages are sent. This network consists of many nodes and enables the communication between users who are at the edge of the network using the LOADng routing protocol. In particular, a single random network topology that includes 400 basic network nodes (neurons) and 80 additional network nodes (microglia). These devices are distributed over an area of 30 km by 24 km on a grid with a side of Δ=1 km. The adopted carrier frequency for this system is $f_{c}=2.45$ GHz, the band is B=5 MHz, the required signal-to-noise ratio and interference is $\theta^{*}=3$ dB, and the transmission power is 2 dBm.

We assume that the connection between any two points may be corrupted with some probability, mainly the probability of a node breakdown is arbitrarily fixed to 0.05, and the mean time to failure is set to 5000 iterations. The simulation time is 10,000 iterations. The average time between messages is 50 iterations. As above, for the sake of clarity, we collect all important simulation parameters used in this part of our work in one table (Table 2).

To deliver data from source to the destination, the well-known LOADng algorithm was proposed. Mainly, at the beginning, the network nodes do not have any information about neighbor nodes. When the necessity of some node discovery occurs, a broadcast message is sent to find specific node. Nodes which are not a destination of such message simply transmit this broadcast message further and the node which is a destination of such message replies with different types of message. Such reply selects a route based on some metric (commonly the distance in the form of hop count) and each node on this route updates its own routing table. In the case of some problems in the existing routing map, e.g., one node does not response, another type of message is sent—which informs all nodes on the route about the problem. The reaction to this message can be turning on microglia-based nodes and dropping routing entities due to such node activation. Then, a new route to the destination needs to be discovered in the same way as described. However, to deal with interference problem in the network, the applied LOADng algorithm was modified. Mainly, the Binary Exponential Backoff algorithm was implemented. In the case of collision (i.e., when at least two transmitters start their transmission at the same time causing harmful interference and degrading effective SINR), the current value of the backoff-window is randomly selected from the allowed range of integer values, and another try for data transmission is repeated after the back-off time is over. Such binary exponential back-off time increases the network reliability, as it introduces some sort of collision mitigation, but it is achieved at the price of longer message processing time. Thus, it increases the average data delivery time within the network, as well as degrades the overall energy consumption.

For performance evaluation, we consider four scenarios:Scenario I: Only neuron-nodes are deployed; this scenario is also used as reference.Scenario II: Both neuron- and microglia-nodes are available.Scenario III: Both neuron- and astrocyte-nodes are deployed.Scenario IV (the complete scenario): All kinds of nodes are deployed (i.e., neuron-nodes, microglia-nodes, and astrocyte-nodes).

As above, neuron-nodes and microglia-nodes are deployed uniformly over a certain geographical areas, where the number of microglia-nodes constitutes 10% of neuron-nodes. If astrocyte-nodes are applied, we assume the presence of one, centrally deployed astrocyte-node in the considered area. Moreover, in our experiments, we assume that a certain level of reliability has to be achieved, and we observed its cost in all scenarios. The structure of the simulator was prepared in such a way that it operates in a slotted mode, mainly the time is split into equal time slots (called iterations). During one iteration, one message may be sent and received by each node. The analysis of the simulation results was carried out in two ways. First, the delay in delivering packets, expressed as the difference between the iteration number in which the message was sent and the iteration number in which the message reached its destination, was analyzed. Second, the analysis of the energy consumption was expressed by the sum of the transmission power in subsequent iterations. In the simplified transmission model, we treat transmitting power as the only component of the consumed power, because it is the largest part of the consumed power in such systems.

In Figure 7, the distribution of the delivery confirmation delay is plotted for the four scenarios. One may observe that inclusion of either microglia- or astrocyte-nodes improves in some ranges the performance when compared with the reference scenario (black line). For example, in Scenario II, there are more situations where very small delivery delays are present. At the same time, the addition of astrocyte-node improves the overall performance significantly. The best results are achieved in general in Scenario III. However, these results cannot be analyzed without association with the overall energy consumption, as shown in Figure 8. Here, one may observe that to some extent Scenarios II–IV are better than the reference one, where only neuron-nodes are considered.

However, the most important conclusion is on network reliability when applying the highly simplified transmission procedures. This analysis has to be done for some specific assumptions; for example, one may request that the maximum allowed delay in the considered networks is less than 100 hops (iterations) or that the total consumed energy during packet transmission is below 100 energy units. Having such constraints allows us to ”slice” vertically the results shown in Figure 7 and Figure 8. For example, if only 100 energy units are allowed, then Scenarios II and III provide the best results. Moreover, for highly stringent scenarios, where, e.g., only 50 hops are allowed, then the reference scenario seems to be the best. It proves that the simplified transmission scenario allows the achievement of a certain reliability level.

## 6. Discussion

In this paper, we analyze the reliability of biology-inspired dense wireless networks. Our considered network is characterized by extreme simplicity of data transmission, i.e., no advanced channel coding is applied. It means that network reliability has to be achieved in a different way. First, we can gain from the fact that the density of deployed nodes is high and in most cases the line-of-sight transmission is possible. It has been proved that such a simplified transmission scenario can achieve reasonable performance expressed in terms of observed message delivery delay or overall energy consumption for a given reliability level. Second, besides the deployment of only simple neuron-nodes (whose basic functionality is to relay messages as only the received signal-to-interference-plus-noise ratio is above some threshold), one may also deploy some nodes with improved functionality.

If we look at the results of the percolation experiment, we can see that it is a very good method for assessing the reliability of a wireless network. For example, when we analyze the impact of the number of nodes on the reliability of the network, we see that increasing the number of nodes (in this case from 30 to 100) significantly lowers the percolation threshold—that is, it is possible to use less reliable devices in this case. However, the analysis of the power of the nodes shows that increasing the power, in fact, increases the range of the nodes, which improves the reliability (as a single node has the chance to connect through more distant nodes). On the other hand, however, we do not observe a significant decrease in the percolation threshold (increase in reliability) because higher power also means a greater level of observed interference—which prevents (or at least limits) reliable transmission. Thanks to the use of percolation, we can analyze the behavior of the wireless network—what is important—regardless of its topology (we assume that it is a random value). In this area of the conducted experiments, it would definitely be worth considering a network where we have the ability to control power levels, which would allow to some extent to manage interference throughout the network. Of course, this approach would improve the performance of the network, but at some cost—increasing the complexity of the entire network. Another aspect worth investigating is the analysis of the radio environment in which there is not always a direct line of sightline.

The analysis of the part concerning the simulation of a specific wireless network shows that, in terms of both—message delay and energy consumption—a network using only basic nodes inspired by neurons performs satisfactorily. Median message delay does not significantly deviate from the minimum values (given the presence of interference in the network); however, some large or extreme delays also occur (e.g., 95th percent delays). In some specific applications or network configurations, however, it is possible to improve the underlying performance of the neurons themselves by applying some kind of support or intelligence in the network (using devices other than neurons as part of the devices). The same situation occurs in the human nervous system where transmission between neurons takes place and is possible; however, it is improved by the use of glial cells. In the case of the simulations carried out, it can be noticed that, for example, the use of additional devices inspired by microglia (scenario two) reduces the lowest network delays so far, however at the same time extending the longest delays in the network—i.e., it is possible to obtain lower delays but at the price of lower delay stability. In general, these simulations have been able to show that replacing some devices with more extensive ones (but not all devices) allows improving or adjusting the behavior of the entire network. As in the case of percolation analysis, it is possible to consider introducing some kind of node control (e.g., power control) here; however, it would also mean increasing the complexity of the network, but it is a promising topic for further research.

Conclusions resulting from the presented results concerning the modeling of a dense wireless network are certainly the possibility of using percolation theory to check the network behavior (in a stochastic way). Secondly, in dense networks, we have the possibility of achieving higher reliability when we apply reduced requirements for data processing (simplification of transmission). This aspect is extremely important in relation to sensor networks where energy consumption is crucial, especially in the case of networks where we cannot replace the battery and each node is limited by the energy reserves it has, and energy harvesting solutions are considered in these networks. In this type of network, the presented simplification of transmission is the most promising. Of course, this requires further research—in particular, the analysis of the impact of the radio environment, the analysis of the aging of devices, and the analysis of failure rates. Device aging and failure frequency are often expressed as reliability versus time, which adds another dimension to the analyses presented in this article.

Summarizing the overall results, one may conclude that, when the dense wireless network is considered, there is a space for the application of highly simplified transmission schemes while keeping the network reliability at an assumed level. Thus, there is a space for further research towards the implementation of other human-nervous-system-focused solutions.

## Figures and Tables

**Figure 1 sensors-21-00576-f001:**
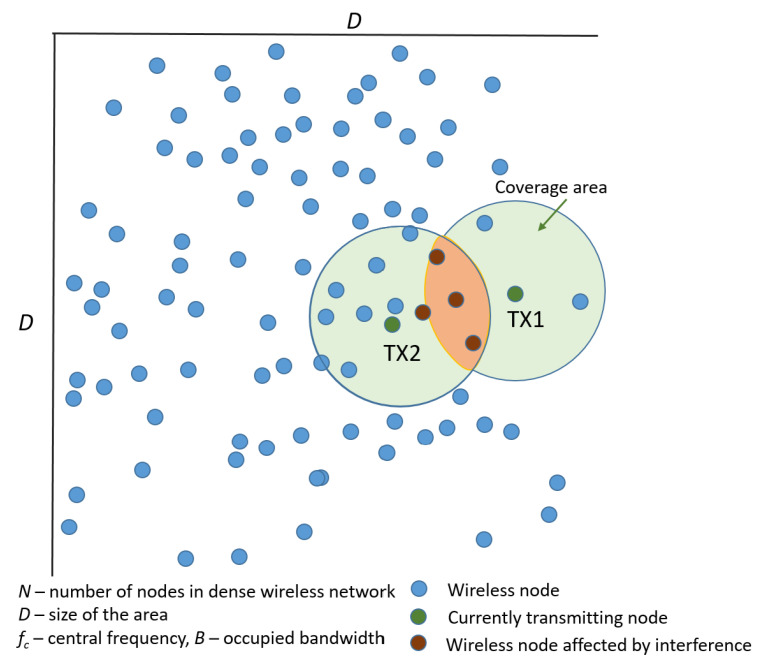
Considered dense wireless network.

**Figure 2 sensors-21-00576-f002:**
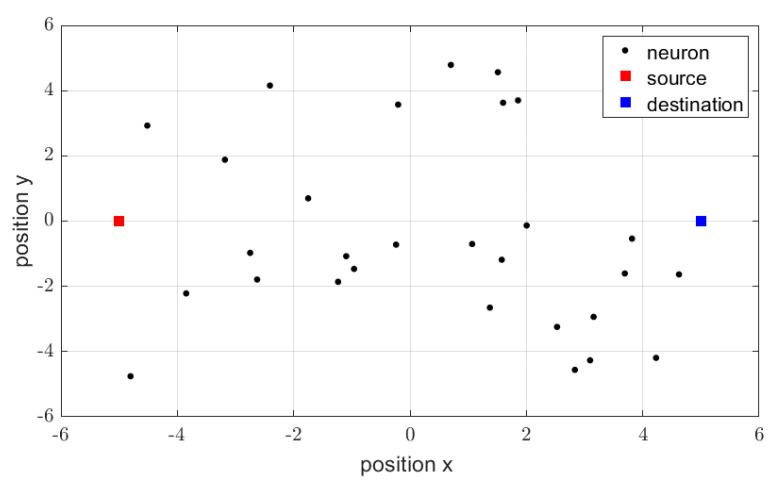
Example of the analyzed topology.

**Figure 3 sensors-21-00576-f003:**
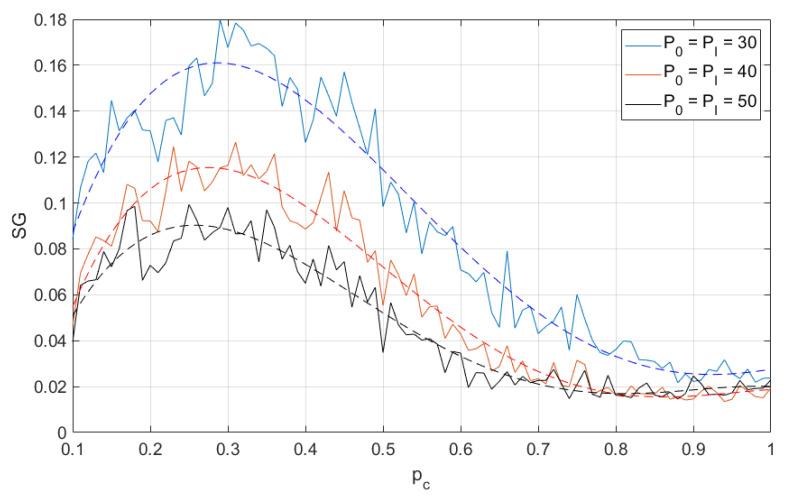
Second largest cluster size as function of node operation probability (different transmit power levels); 30 neuron-nodes. Dashed lines are a fourth-degree polynomial approximation.

**Figure 4 sensors-21-00576-f004:**
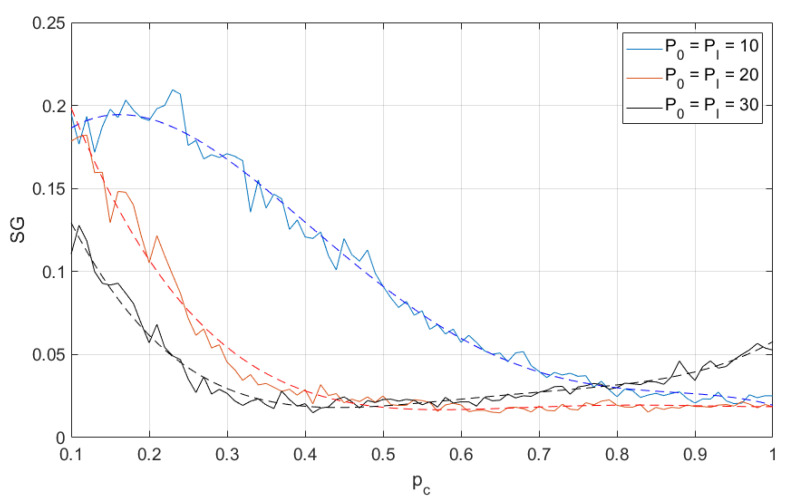
Second largest cluster size as function of node operation probability (different transmit power levels); 100 neuron-nodes. Dashed lines are a fourth-degree polynomial approximation.

**Figure 5 sensors-21-00576-f005:**
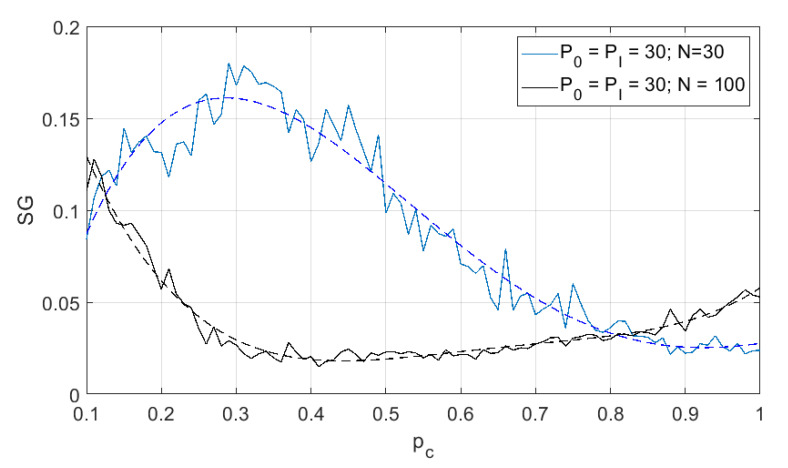
Second largest cluster size as function of node operation probability (different number of nodes). Dashed lines are a fourth-degree polynomial approximation.

**Figure 6 sensors-21-00576-f006:**
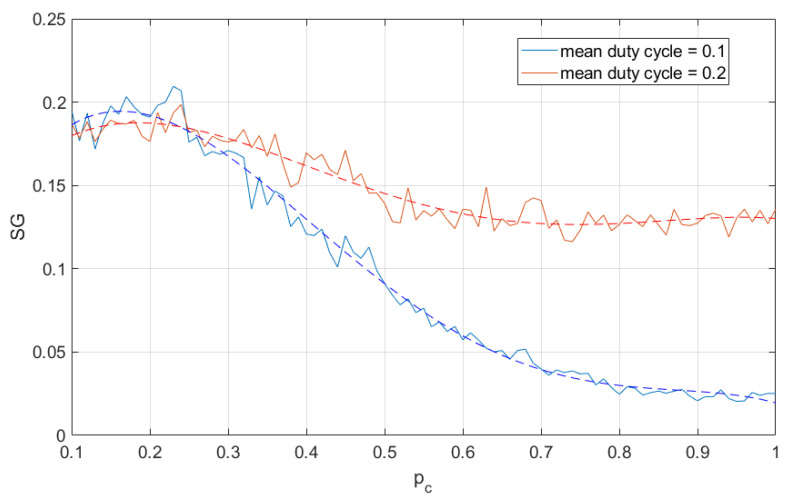
Second largest cluster size as function of node operation probability (different mean duty cycle). Dashed lines are a fourth-degree polynomial approximation.

**Figure 7 sensors-21-00576-f007:**
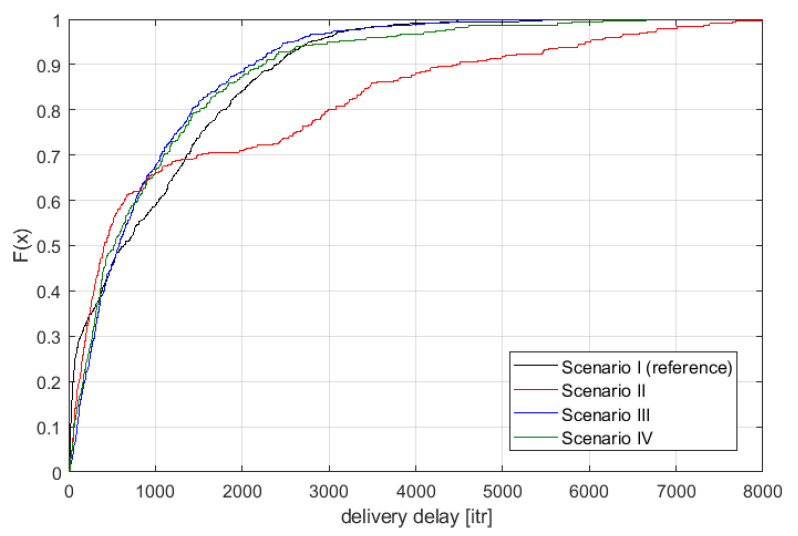
Distribution of delivery confirmation delay.

**Figure 8 sensors-21-00576-f008:**
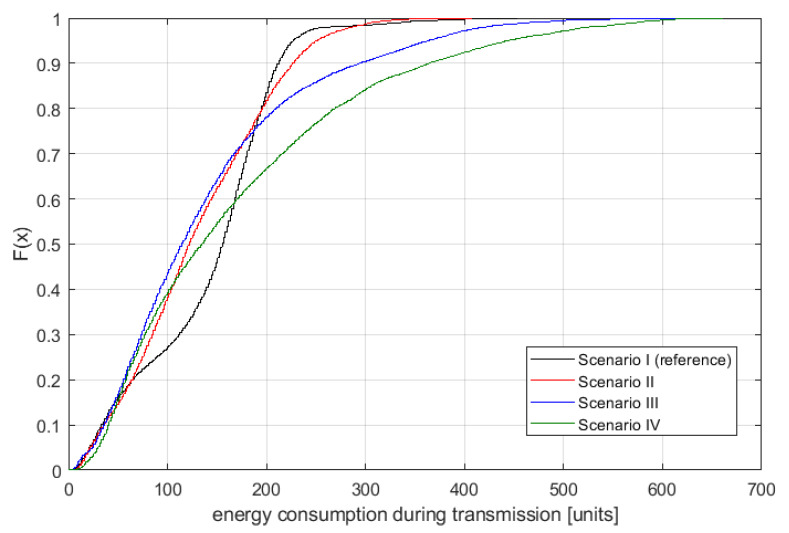
Distribution of energy consumption during transmission.

**Table 1 sensors-21-00576-t001:** Simulation parameters used in the first phase of the study.

Parameter	Description	Value
*N*	number of the neuron nodes	30
*M*	number of the microglia nodes	M=0.1·N=3
*D*	size of the considered area (related to the cell range size)	10 [distance units]
*r*	transmission range (cell range)	1 [distance unit]
$P_{0}=P_{I}$	transmit power of any node	1, 10, 30, 40 or 50 [power units]
$\sigma^{2}$	noise power	1 [power units]
$\theta^{*}$	required SINR level	1 [linear scale] or 0 [dB]

**Table 2 sensors-21-00576-t002:** Simulation parameters used in the second phase of the work.

Parameter	Description	Value
*N*	number of the neuron nodes	400
*M*	number of the microglia nodes	80
*A*	number of the astrocyte nodes	1
x×y	sizes of the considered area	30×24 km
Δ	grid size	1 km
$f_{c}$	center frequency	2.45 GHz
*B*	frequency bandwidth	5 MHz
$P_{0}=P_{I}$	transmission power	2 dBm
$\theta^{*}$	SINR threshold	3 dB

## Data Availability

Not applicable.

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
