# Peer review of "Brain-Inspired Data Transmission in Dense Wireless Network"

_sensors, 2021, doi:10.3390/s21020576_

Round 1

Reviewer 1 Report

This paper is focused on an interesting topic, related with wireless network planning, based on (or even inspired by) the human nervous system. It is pleasant to notice that more and more scientific disciplines and fields of study tend to draw ideas and inspiration from biology. Indeed, similarities are not mere coincidence. Many ICT applications could benefit from it.

Authors precisely point out how important is to effectively and efficiently manage bandwidth resources, under limited network conditions. Nowadays, thanks to the widespread of mobile technologies, communication standards, consumer devices, etc., each and every way of transmitting data seems undervalued and underinvested. Hopefully, new solutions will come with the emergence of 5G and future generation standards. Authors propose and describe such a solution.

The displayed graphs and plots are clear and easily understandable. Discussed mathematical formulas and equations seem plausible and free of error. Symbols are written in italics in order to distinguish them from plain text.

Personally, I would like to see a Related Work section and/or more information in the Discussion section. Additionally, Authors could provide information about the laboratory stand, including utilized simulation software (environment). Data considering e.g. used libraries or data sets is always most welcomed. Custom solutions are always highly esteemed. This could be a starting point for discussions with other Authors.

Authors entitled this paper as [… Highly Simplified…]. Is this simplification related with: network signaling, single- or multi-path transmission, simple (e.g. small, primary) or complex (e.g. big, multimedia) data delivery, or complexity (e.g. energy efficiency, implementation costs, potential collisions, real-time operation)? Consider extending the description of your approach, e.g. by comparing your solution with other ones.

Minor editorial issues:

  • Abbreviations – e.g. LOS (Line-of-Sight), SINR/SNIR (…) – each one should be provided in both full and short form, when first appearing in text.
  • Unnecessary (or double) space signs – on numerous pages.
  • Missing words – e.g. such [as] glucose… have been already widely [known/applied] in communications…
  • Using other fonts and/or size – e.g. [neuron-nodes], [microglia-nodes], [astrocyte-node], as well as [four nines], [reliability], etc.
  • Using the plural – e.g. [will run in a different way] or [will run in different ways].

Carefully examine and double-check the whole text.

Overall, the manuscript is written in proper-English, it is pleasant to read.

In my opinion, this paper could be accepted for publication. However, in its current form, it requires a major revision.

The Abstract is informative, it adequately describes the content of the manuscript. However, in my opinion, the title should reflect the content of the paper more accurately. I would advise the Authors to add [Brain-Inspired] or [Glial Cell Inspired], etc., resulting in e.g. [Highly Simplified Brain-Inspired Data Transmission (Method/Scheme) in Dense Wireless Network].

When examining available literature on the aforementioned topic, together with those cited in the References section, I have encountered a number of publications. They include, inter alia, the Authors’ own work. However, not all papers are cited in text.

It is good that the Authors carry out their investigation over a long period of time, and continue to publish novel results. Nonetheless, significant publications are missing, including those recently published in 2019-2020 and available in Open Access:

  • Journal of Telecommunications and Information Technology – Neuroplasticity and Microglia Functions Applied in Dense Wireless Networks;
  • International Journal of Electronics and Telecommunications – Dense Wireless Network Inspired by the Nervous System;

as well as conference papers from: SoftCOM, WoWMoM, MSN, PIMRC.

I am not a supporter of excessive self-citation. However, I believe that the most important (due to research progress, etc.) works should be included in the References section, indicating key changes (e.g. milestones) and research developments. Some portions of text look too similar, some figures look too similar as well. Authors should clearly point out the novelty and highlight new content presented in this paper. This information would be helpful to many potential readers, including researchers as well as academics.

Reviewer 2 Report

I would like to point out that the article is interesting, it relates to network structures of great importance and applications for IoT. However, it requires numerous corrections and additions.

The authors should describe the current state of knowledge on the use of inspiration from biological systems in network modeling, including dense wireless networks.

The authors should complete the information on the simulation environment and the assumed initial parameter values.

There is no information on how delivery confirmation delay and energy consumption were calculated? What assumptions are made for these calculations? Were all nodes consuming energy the same way?

Reading this paper, one has the impression that the authors used the same environment and the same test scenarios as in the previous paper. Therefore, they should very strongly indicate what is completely new, how the performance of the network has improved compared to the previous model.

What do the authors consider a new contribution compared to papers [13] and [16]?

Moreover, there is no definition of the function of the probability. This is related to the lack of a clear transition from sections 3.1 to 3.2.

The summary is too general. The authors should strongly emphasize the impact of the obtained results on modeling the operation of dense wireless networks.

Round 2

Reviewer 1 Report

Thank you for your time and quick response. In my opinion, the new, extended version of the manuscript has improved the quality of the paper.

Authors responded correctly to my questions and suggestions. All doubts were explained in detail in the sent cover letter. Relevant parts of the manuscript have been expanded and/or modified accordingly. I hereby declare that in my opinion the paper fulfills all requirements. I do recommend it to be published in Applied Sciences.

Merry Christmas and a Happy New Year!

Author Response

We thank you for your positive answers and wishes! We hope that you had also a serene Christmas time and New Year!  We wish you all the best in the just started 2021 year.

Reviewer 2 Report

The authors dispelled all my doubts. It is a pity that, when preparing the initial version of the manuscript, they did not pay attention to its shortcomings. At the moment, I have no additional comments to the article. For the future, I suggest that you carefully analyze the article before submitting it, especially in the context of the thematically related cycle of previous papers.

Author Response

We thank you for your positive comments, as well as for your valuable suggestions on paper preparation. We wish you all the best in the New Year.